# First-line glaucoma monotherapy medication patterns in Finland during 1995–2019 based on a population-based study

**Petri K. M. Purola**[1,2]*, **Seppo V. P. Koskinen**[3], **Hannu M. T. Uusitalo**[1,2,4]

**1** Faculty of Medicine and Health Technology, Department of Ophthalmology, Tampere University, Tampere, Finland, **2** Finnish Register of Visual Impairment, Finnish Federation of the Visually Impaired, Helsinki, Finland, **3** Department of Public Health and Welfare, Finnish Institute for Health and Welfare, Helsinki, Finland, **4** Tays Eye Center, Tampere University Hospital, Tampere, Finland

* petri.purola@tuni.fi

**Data Availability Statement:** The data from FinHealth 2017 and its precursors Health 2000 and Health 2011 are not publicly available as they include confidential information that could

## Abstract

### Background

The long-term patterns in first-line glaucoma medication are not well established. Exploring these in longitudinal and population-based settings would provide information for the health-care systems to plan glaucoma care accordingly.

### Objective

To evaluate patterns in first-line glaucoma monotherapy in Finland during 1995–2019 based on nationwide survey and register data.

### Methods

A population-based cohort study with 25 years of total follow-up. The cohort ($n = 9288$) is a random sample drawn from the nationwide health examination survey FinHealth 2017 which represents the Finnish population aged 30 years or older in 2017. Glaucoma patients were selected from the survey participants based on linked register data that included prescriptions and special reimbursements for glaucoma medication. The patterns, length of use, age at initiation, and persistence of first-line glaucoma drug monotherapies were observed during 1995–2019.

### Results

A total of 141 glaucoma patients with glaucoma drug monotherapy as a first-line glaucoma therapy were identified from the sample. The proportion of patients continuing with their first-line monotherapy was 64% after 1 year, 37% after 3 years, and 21% after 5 years of follow-up. During the 25 years there was a shift from beta-blockers to prostaglandin analogues as the prominent first-line glaucoma drug class. The length of use was longer for prostaglandin analogues compared with beta-blockers among patients continuing with their first-line monotherapy after 5 years of follow-up. The non-persistence rate was 38% of all patients

compromise the privacy of the participants. The data can be used for research and monitoring of health, wellbeing, functioning, and use of services of the population at THL and with collaborators based on collaboration agreement (more information: terveys-2000-2011@thl.fi). The data available from the THL Biobank cover those who have participated in the health examination and have given consent to biobanking and can be applied via the THL Biobank in accordance with the biobank act and THL Biobank research areas (thl.fi/biobank). The Finnish Social and Health Data Permit Authority Findata may grant permits in accordance with the act on the secondary use of social and health data in Finland (www.findata.fi/en). Further inquiries can be directed to the email address terveys-2000-2011@thl.fi.

**Funding:** PKMP has received research grants from Tampereen seudun Näkövammaisten tukisäätiö s.r, Tampere, Finland (https://www.tsnvtukisaatio.fi); Elsemay Björn Fund, Helsinki, Finland (URL not available); Finnish Federation of the Visually Impaired, Helsinki, Finland (https://www.nkl.fi); Emil Aaltosen Säätiö, Tampere, Finland (https://emilaaltonen.fi); and Juho Vainion Säätiö, Helsinki, Finland (www.juhovainionsaatio.fi). The funders had no role in study design, data collection and analysis, decision to publish, or preparation of the manuscript.

**Competing interests:** The authors have declared that no competing interests exist.

during their first-line monotherapy. Timolol fixed-combinations were the most common second-line glaucoma therapy with a share of 39% after 5 years of follow-up.

## Conclusions

During the 25-year follow-up a shift from beta-blockers to prostaglandin analogues had occurred and long initial therapies of over 5 years had become more common. However, the decline in the continuation of the initial therapy still occurred early with 1 out of 3 patients continuing after 3 years. This decline together with the consistent problem of non-persistence remain clinical challenges in topical drug therapy of glaucoma.

## Introduction

Glaucoma is a group of optic neuropathies that can lead to irreversible blindness if left untreated. Elevated intraocular pressure (IOP) is considered the modifiable risk factor for glaucoma [1]. Therefore, the current treatment of glaucoma aims to delay the progression of the disease and consequent vision loss by lowering the IOP via medication, laser treatments, or surgery [1,2].

The treatment of glaucoma is usually initiated with topical antihypertensives and recommended to be initiated as a monotherapy but can be switched to another monotherapy or a combination drug therapy with two or more IOP-lowering agents in case the current therapy is deemed insufficient [3,4]. Alternatively, laser treatments such as laser trabeculoplasty and laser iridotomy or surgery can be offered as a first-line glaucoma therapy, in patients with drug-related adverse reactions, in patients with difficulties in administering topical medication, and in patients with severe disease.

Glaucoma treatment requires lifelong and frequent follow-up. The adherence to the recommended medication among glaucoma patients is a key challenge for treatment success in glaucoma. Numerous studies have demonstrated that low adherence and persistence with topical and systemic glaucoma treatment is common [5–8], which can manifest as progression of vision loss and eventual blindness [9].

However, the long-term patterns in the first-line glaucoma medication are not well established. Previous studies on the subject have been limited to short, less than 10-year follow-up periods [8,10–13] and samples that are not representative at the population level [12,14]. Exploring these patterns in longitudinal and population-based settings with comprehensive follow-up periods would provide the highly needed information for the healthcare systems to plan glaucoma care accordingly. Therefore, in the present study our aim was to analyze the patterns in the first-line glaucoma medication during a total follow-up period of 25 years based on a population-representative cohort of Finnish adults from a nationwide health examination survey complemented with national health register data on glaucoma and glaucoma medication.

## Materials and methods

### Study design and data

This study was based on FinHealth 2017, a cross-sectional nationwide health examination survey conducted in 2017 by the Finnish Institute for Health and Welfare (THL) [15]. The goal of the survey was to collect comprehensive, up-to-date information on health, functional

capacity, and welfare of Finnish adults. The survey included comprehensive self-reported assessments as well as a health examination conducted at a nearby screening center.

The survey included a nationally representative sample of 10,247 Finnish adults aged 18 years or older with a participation rate of 68.8% for participating at any phase of data collection. The details of the design and sampling of the survey have been described previously [16]. To summarize, the sample was drawn using two-stage stratified cluster sampling, and persons aged 80 or older were oversampled by doubling the sampling fraction. The sample weights were calibrated by post-stratification defined by age, sex, region, and native language to account for non-response and missing data. Separate weights were applied for the survey to produce results representing the Finnish adult population in 2017. For the present study we selected 9,288 participants aged 30 years or older as the study cohort.

The survey sample was linked to the Social Insurance Institution of Finland (KELA) registers to obtain data on the special reimbursement for glaucoma medication during 1972–2019 and the number of glaucoma medication prescriptions during 1995–2019. Data on glaucoma-related diagnoses and operations were obtained from the Care Registers for Social Welfare and Health Care based on inpatient care (HILMO) during 1996–2019 as well as glaucoma-related diagnoses based on specialized health care outpatient visits (AvoHILMO) during 2011–2019. The HILMO and AvoHILMO data are collected automatically from healthcare service providers' patient management systems and delivered to THL.

## Ethics approval and informed consent

The current study was conducted in line with the tenets of the Helsinki Declaration. All procedures in the FinHealth 2017 Survey involving human participants were performed in accordance with the ethical standards of the institutional and national research committees, and with the 1964 Helsinki declaration and its later amendments or comparable ethical standards. The FinHealth 2017 Survey was approved by the Coordinating Ethics Committee at the Hospital District of Helsinki and Uusimaa with reference 37/13/02/00/2016. All participants received an information letter regarding the study beforehand. Written informed consent was obtained from all participants. The ethical approval process details have been reported in a previous publication [16].

## Glaucoma patient selection

We classified survey participants into glaucoma patients according to the following register-based conditions: 1) entitlement to special reimbursement for glaucoma medication based on KELA data during 1972–2019; or 2) glaucoma diagnosis (International Classification of Diseases diagnosis codes H40 and H40.1–H40.9 for version 10) based on HILMO data during 1996–2019 and AvoHILMO data during 2011–2019; or 3) glaucoma-related eye operation (NOMESCO Classification for Surgical Procedures CHB50, CHD00, CHD05, CHD15, CHD50, CHD60, CHD65, CHD99, and CHF05) based on HILMO data during 1996–2019.

## Glaucoma medication data

In Finland, glaucoma medication is distributed by pharmacies every 3 months, and the data on the number and type of used glaucoma medications during these 3-month periods were available for all study participants. We classified glaucoma drugs based on glaucoma medication prescriptions and refills with ATC code S01E* obtained from KELA during 1995–2019. To allow the scrutinization of first-line glaucoma drugs we included only those with glaucoma medication initiated with drug monotherapy during 1995–2019.

The age at initiation of therapy was based on the date of the first prescription. We calculated the length of use as the time between the date of the first prescription of the first drug and 1) the date of the first glaucoma-related eye operation; or 2) the date of the first prescription of the second drug if preceding a possible glaucoma-related eye operation; or 3) the last refill of the first drug if it was not switched during the follow-up.

The analyses were based on both the drug class and its type. The classes (and types) included beta-blockers (betaxolol, timolol), alpha-2 agonists (brimonidine), prostaglandin analogues (bimatoprost, latanoprost, tafluprost, travoprost), carbonic anhydrase inhibitors (brinzolamide, dorzolamide), and others (apraclonidine, pilocarpine).

## Statistical analyses

All analyses were carried out using R software (v. 4.3.1, R Core Team, R Foundation for Statistical Computing, Austria). We accounted the sampling design of the survey for by using Survey package 4.2–1 for R [17] and a weighting scheme calculated by THL. We expressed the results as means, standard deviations (SDs), and 95% confidence intervals (CIs). We created a Kaplan–Meier estimator and tested it with log-rank and Gehan–Wilcoxon tests using package survival 3.5–7 [18,19]. We drew a Sankey diagram using package PantaRhei 0.1.2 [20]. The distribution of the analyzed data was skewed according to Shapiro–Wilk normality test. Therefore, we used Mann–Whitney U test for analyzing the mean age at initiation of therapy of the two most common first-line glaucoma drug classes and Kruskal–Wallis test for the analysis of the five most common drug types adjusted with Dunn–Bonferroni correction using function DunnTest from package DescTools 0.99.50 [21]. There were no outliers. All p-values were two-sided and were considered statistically significant when less than 0.05.

## Data availability

The data from FinHealth 2017 and its precursors Health 2000 and Health 2011 are not publicly available as they include confidential information that could compromise the privacy of the participants. The data can be used for research and monitoring of health, wellbeing, functioning, and use of services of the population at THL and with collaborators based on collaboration agreement (more information: terveys-2000-2011@thl.fi). The data available from the THL Biobank cover those who have participated in the health examination and have given consent to biobanking and can be applied via the THL Biobank in accordance with the biobank act and THL Biobank research areas (thl.fi/biobank). The Finnish Social and Health Data Permit Authority Findata may grant permits in accordance with the act on the secondary use of social and health data in Finland (www.findata.fi/en). Further inquiries can be directed to the email address terveys-2000-2011@thl.fi.

## Results

Out of the total 9,288 participants aged 30 years or older in the FinHealth 2017 Survey, 6,425 (69.2%) had partaken in at least one part of the survey. The selection process of the analyzed participants from this eligible sample is illustrated in Fig 1. We observed a total of 223 subjects with register-based glaucoma with a prevalence of 3.1% (95% CI 2.7–3.6) in the population aged 30 years or older. Out of the 223 glaucoma patients 203 were treated with medication, of which 62 were excluded: 43 (21%, 43/203) started with multitherapy as a first-line glaucoma therapy, 4 had existing multitherapy since start of the follow-up in 1995, 12 were treated with laser therapy or surgery as a first-line glaucoma therapy, and 3 had only a single glaucoma medication prescription during the follow-up. We selected a total of 141 (69%, 141/203) glaucoma patients with drug monotherapy as a first-line glaucoma therapy for the analyses with a

**Fig 1. Flowchart of the selection of glaucoma patients treated with first-line monotherapy in the FinHealth 2017 survey sample representing the Finnish adult population aged 30 years or older.** SD = standard deviation.

prevalence of 2.0% (95% CI 1.6–2.3) in the population aged 30 years or older. The distribution of glaucoma subtypes among these analyzed subjects is shown in S1 Table, with primary/ chronic open-angle glaucoma being the most prevalent diagnosis (41%) followed by exfoliative glaucoma (17%). During the 25 years of total follow-up 59% (*n* = 83) of the analyzed participants switched to another drug, 18% (*n* = 26) had no change in their initial therapy during the

follow-up, 16% ($n$ = 23) had a second drug added alongside the first-line drug, and 6% ($n$ = 9) were treated with laser therapy or cataract surgery as a second-line therapy.

The distribution of first-line glaucoma drug classes and types in the survey sample is shown in Table 1. The proportion of patients continuing with their first-line glaucoma monotherapy after 1, 3, and 5 years of follow-up is shown in Table 2. The proportion of patients continuing with their initial monotherapy among all first-line glaucoma drugs was 64% (90/141) after 1 year, 37% (52/141) after 3 years, and 21% (29/141) after 5 years of follow-up. Prostaglandin analogues were the most common first-line glaucoma drug class with a share of 70% (98/141) followed by beta-blockers with 23% (33/141). Latanoprost was the most common first-line glaucoma drug type with a share of 38% followed by timolol with 20%, tafluprost with 13%, travoprost with 12%, and bimatoprost with 6%.

The non-persistence regarding all first-line glaucoma monotherapies as well as prostaglandin analogues and beta-blockers specifically is shown in Table 3 and Fig 2. Overall, 62% of all patients did not have significant delays (refills delayed < 1 months) between their refills during their first-line monotherapy, while 38% had at least 1 refill delayed $\geq$ 1 months and 20% at least 1 refill delayed $\geq$ 3 months. The persistence did not differ noticeably between prostaglandin analogues and beta-blockers. During the 25-year follow-up the proportion of delays of $1 \geq$ months decreased while the proportion of delays of $3 \geq$ months increased. The persistence for the other drug classes and types in specific was not determined due to low number of participants ($n \leq 5$).

The time trends in the two most common first-line glaucoma drugs classes, first-line (selective) laser trabeculoplasty, and the most common second-line therapy of the cohort, timolol fixed-combinations, during 1995–2019 are presented in Fig 3. Between 1997 and 2002 beta-blockers were the most common first-line glaucoma drug class. The use of prostaglandin analogues increased during 1999–2002, and from 2003 onwards they have remained as the most prevalent first-line glaucoma drug class.

**Table 1. Number and proportion of glaucoma patients in the FinHealth 2017 Survey sample representing the Finnish adult population aged 30 years or older according to their first-line glaucoma monotherapy drug class and type during the follow-up between 1995 and 2019.**

| First-line glaucoma drug class and type | n and % |
|---|---|
| Prostaglandin analogue | 98 (70) |
| *Latanoprost* | 54 (38) |
| *Tafluprost* | 19 (13) |
| *Travoprost* | 17 (12) |
| *Bimatoprost* | 8 (6) |
| Beta-blocker | 33 (23) |
| *Timolol* | 28 (20) |
| *Betaxolol* | 5 (4) |
| Carbonic anhydrase inhibitor | 4 (3) |
| *Brinzolamide* | 3 (2) |
| *Dorzolamide* | 1 (1) |
| Alpha-2 agonist | 3 (2) |
| *Brimonidine* | 3 (2) |
| Other | 3 (2) |
| *Pilocarpine* | 2 (1) |
| *Apraclonidine* | 1 (1) |
| Total | 141 (100) |

**Table 2. Number and proportion of glaucoma patients continuing with their first-line therapy after 1, 3, and 5 years of follow-up in the FinHealth 2017 Survey sample representing the Finnish adult population aged 30 years or older according to their first-line glaucoma monotherapy drug class and type during the follow-up between 1995 and 2019.**

| First-line glaucoma drug class and type | n at the start | n and % continuing after 1 year | n and % continuing after 3 years | n and % continuing after 5 years |
|---|---|---|---|---|
| Prostaglandin analogue | 98 | 61 (62) | 34 (35) | 19 (19) |
| *Latanoprost* | 54 | 36 (67) | 19 (35) | 13 (24) |
| *Tafluprost* | 19 | 9 (47) | 5 (26) | 3 (16) |
| *Travoprost* | 17 | 11 (65) | 8 (47) | 2 (12) |
| *Bimatoprost* | 8 | 5 (63) | 2 (25) | 1 (13) |
| Beta-blocker | 33 | 24 (73) | 15 (46) | 9 (27) |
| *Timolol* | 28 | 22 (79) | 13 (46) | 7 (25) |
| *Betaxolol* | 5 | 2 (40) | 2 (40) | 2 (40) |
| Carbonic anhydrase inhibitor | 4 | 2 (50) | 2 (50) | 0 (0) |
| *Brinzolamide* | 3 | 2 (67) | 2 (67) | 0 (0) |
| *Dorzolamide* | 1 | 0 (0) | 0 (0) | 0 (0) |
| Alpha-2 agonist | 3 | 1 (33) | 0 (0) | 0 (0) |
| *Brimonidine* | 3 | 1 (33) | 0 (0) | 0 (0) |
| Other | 3 | 2 (67) | 1 (33) | 1 (33) |
| *Pilocarpine* | 2 | 1 (50) | 1 (50) | 1 (50) |
| *Apraclonidine* | 1 | 1 (100) | 0 (0) | 0 (0) |
| Total | 141 | 90 (64) | 52 (37) | 29 (21) |

The length of use of the two most common first-line glaucoma drug classes and the five most common drug types during 1995–2019 is illustrated in Fig 4 as survival curves. There were no statistically significant differences in the length of use were between the two glaucoma drug classes and the five glaucoma drug types during the total follow-up time. However, after 5 years of follow-up the length of use was statistically significantly longer for prostaglandin analogues compared with beta-blockers according to the log-rank test (p = 0.02). After 7 years of follow-up this difference was significant also according to the Gehan–Wilcoxon test (p = 0.02). We could not determine the length of use for the other drug classes and types due to low number of participants ($n \leq 5$).

The mean age at initiation of therapy of the two most common first-line glaucoma drug classes and the five most common drug types during 1995–2019 is illustrated in Fig 5. The mean age at initiation was significantly (p = 0.002) lower for beta-blockers with a mean of 59.9 years (95% CI 56.3–63.4) in comparison to prostaglandin analogues with a mean of 66.9 years (95% CI 64.6–69.1). In relation to this, post hoc pairwise tests determined that timolol, with a mean of 59.6 years (95% CI 56.0–63.1) was associated with the lowest age at initiation in

**Table 3. Non-persistence regarding all first-line glaucoma monotherapies and the two most common glaucoma drug classes during the follow-up between 1995 and 2019 in the FinHealth 2017 Survey sample representing the Finnish adult population aged 30 years or older.**

| First-line glaucoma drug class | Patients treated with first-line monotherapy | Patients with at least 1 refill delayed ≥ 1 months | Patients with at least 1 refill delayed ≥ 3 months | Refills | Refills delayed ≥ 1 months | Refills delayed ≥ 3 months |
|---|---|---|---|---|---|---|
| Prostaglandin analogue | 98 | 37 (38%) | 20 (20%) | 965 | 117 (12%) | 44 (5%) |
| Beta-blocker | 33 | 14 (42%) | 6 (18%) | 304 | 50 (17%) | 6 (2%) |
| All | 141 | 54 (38%) | 28 (20%) | 1381 | 172 (12%) | 52 (4%) |

Other first-line glaucoma drug classes are not listed due to low number of participants.

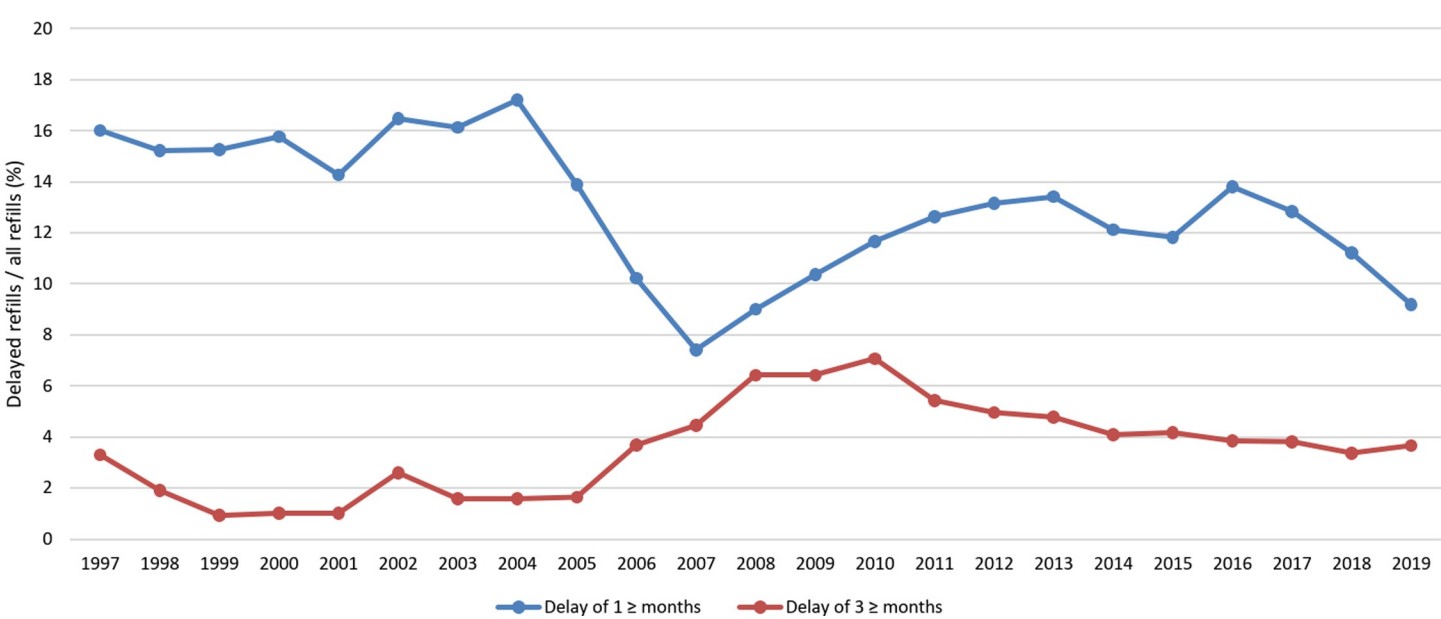

**Fig 2. Non-persistence in first-line glaucoma monotherapy during the follow-up between 1995 and 2019 in the FinHealth 2017 Survey sample representing the Finnish adult population aged 30 years or older.** Numbers were smoothed using a 3-year central moving average starting from 1997 with a 2-year wash-out period included at the start of the follow-up.

comparison to those of the four other common drugs (p = 0.005), which were 65.5 years (95% CI 62.5–68.6) for latanoprost, 69.4 years (95% CI 65.0–73.8) for tafluprost, 68.1 years (95% CI 63.7–72.6) for travoprost, and 66.6 years (95% CI 61.3–71.8) for bimatoprost. This difference is likely explained by the fact that the use of timolol was particularly common during 1998–

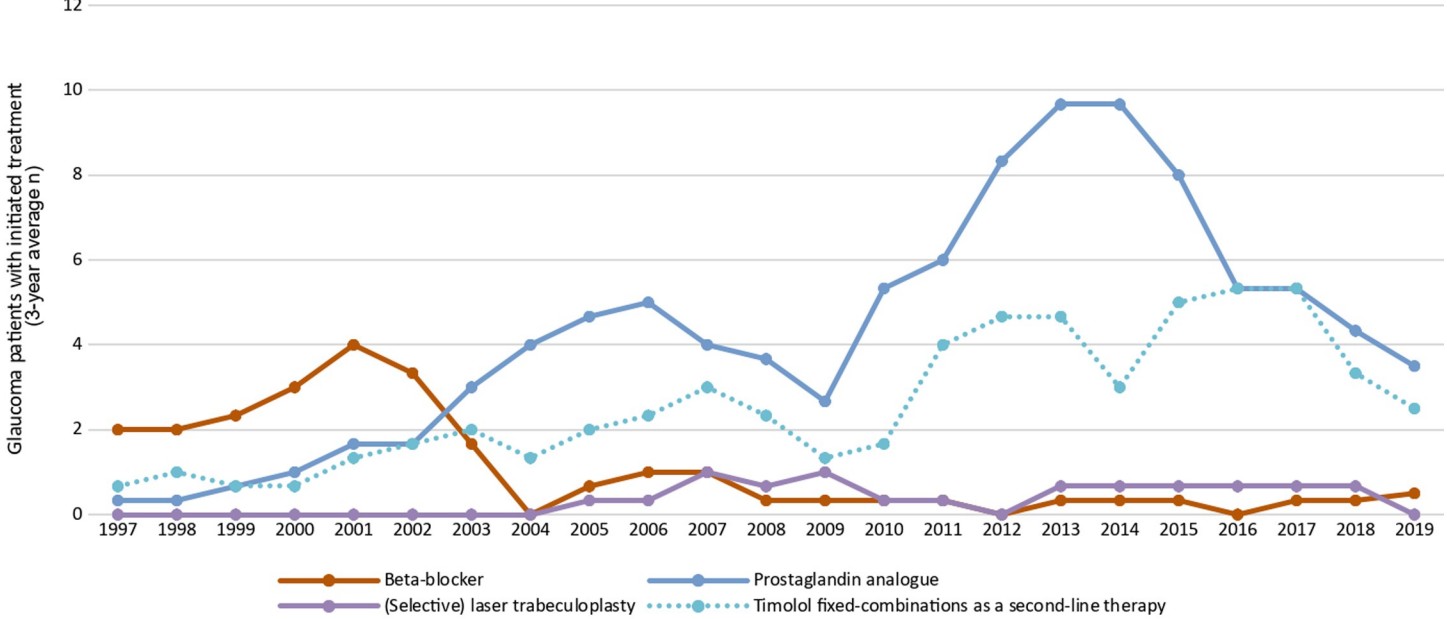

**Fig 3. Time trends in glaucoma treatment during the follow-up between 1995 and 2019 in the FinHealth 2017 Survey sample representing the Finnish adult population aged 30 years or older.** Numbers were smoothed using a 3-year central moving average starting from 1997 with a 2-year wash-out period included at the start of the follow-up.

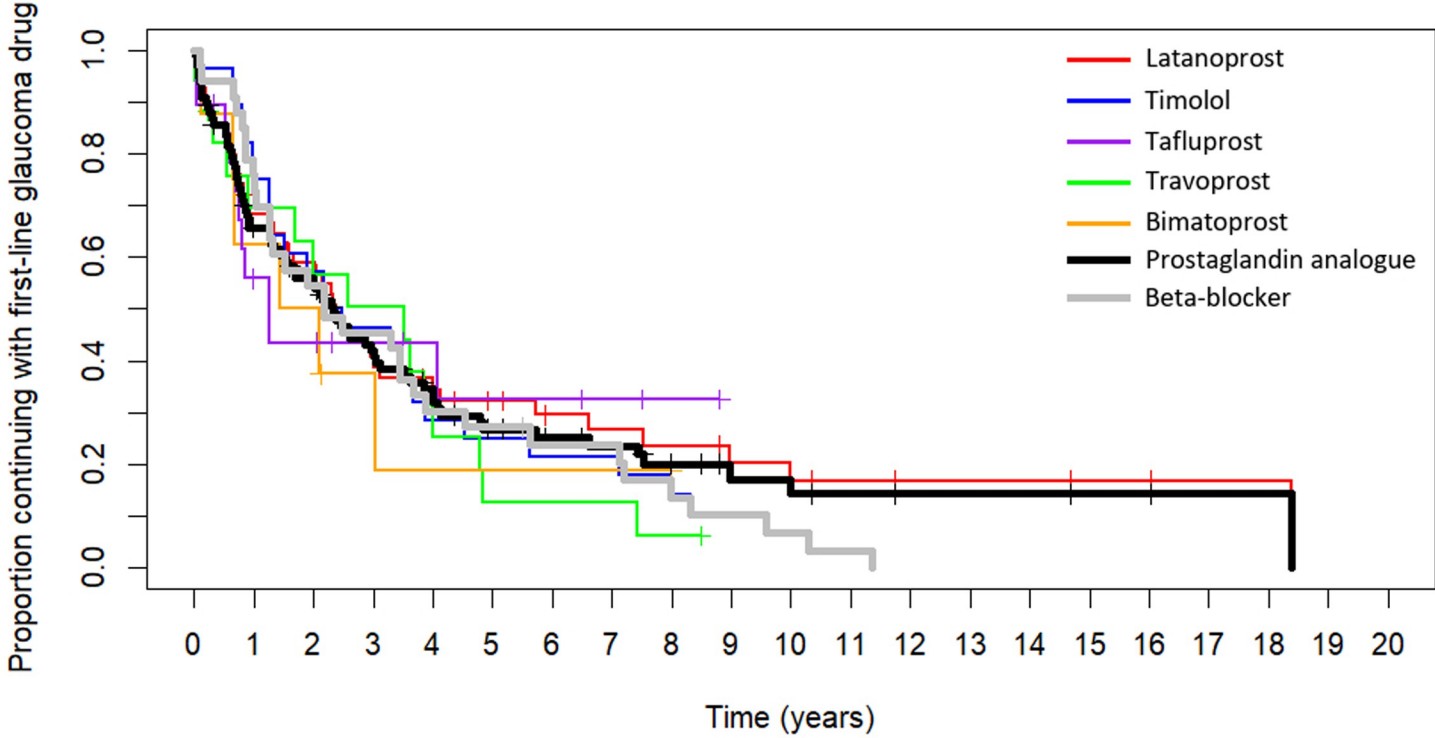

**Fig 4. Plot of Kaplan–Meier estimator on the length of use of the two most common first-line glaucoma monotherapy drug classes and the five most common drug types during the follow-up between 1995 and 2019 in the FinHealth 2017 Survey sample representing the Finnish adult population aged 30 years or older.** Participants without switch to second-line therapy during the follow-up were considered right-censored with their endpoints shown as vertical lines. After 5 years of follow-up there was a significant difference in the length of use between prostaglandin analogues and beta-blockers according to the log-rank test (p = 0.02), and after 7 years the difference was significant also according to the Gehan–Wilcoxon test (p = 0.02).

2002 when the 2017 cohort was 15–19 years younger (Fig 3). We could not determine the mean age at initiation for the other drug classes and types due to low number of subjects ($n \leq 5$).

The patterns in switching from the five most common first-line glaucoma drug types to second-line therapies during 1, 3, and 5 years of follow-up and during total follow-up in 1995–2019 are illustrated by Sankey diagrams in Fig 6. The proportions of patients continuing with their first-line glaucoma drug after 1, 3, and 5 years of follow-up and total follow-up were 72%, 41%, 33%, and 24% for latanoprost; 81%, 50%, 29%, and 0% for timolol; 53%, 38%, 38%, and 42% for tafluprost; 69%, 53%, 14%, and 12% for travoprost; and 63%, 29%, 17%, and 25% for bimatoprost, respectively. During 1 year of follow-up 9% switched to another monotherapy, 17% switched to timolol fixed-combinations, and 3% switched to (selective) laser trabeculoplasty. During 3 years of follow-up 20% switched to another monotherapy, 32% switched to timolol fixed-combinations, and 5% switched to (selective) laser trabeculoplasty. During 5 years of follow-up 27% switched to another monotherapy, 39% switched to timolol fixed-combinations, and 5% switched to (selective) laser trabeculoplasty. And during the total follow-up 29% switched to another monotherapy, 44% switched to timolol fixed-combinations, and 4% switched to (selective) laser trabeculoplasty. Timolol fixed-combinations were the most common second-line therapy with a share of 17% (20/118), 32% (34/107), and 39% (36/92) after 1, 3, and 5 years of follow-up, respectively, and its share was 44% (55/126) of all second-line therapies used during the total follow-up.

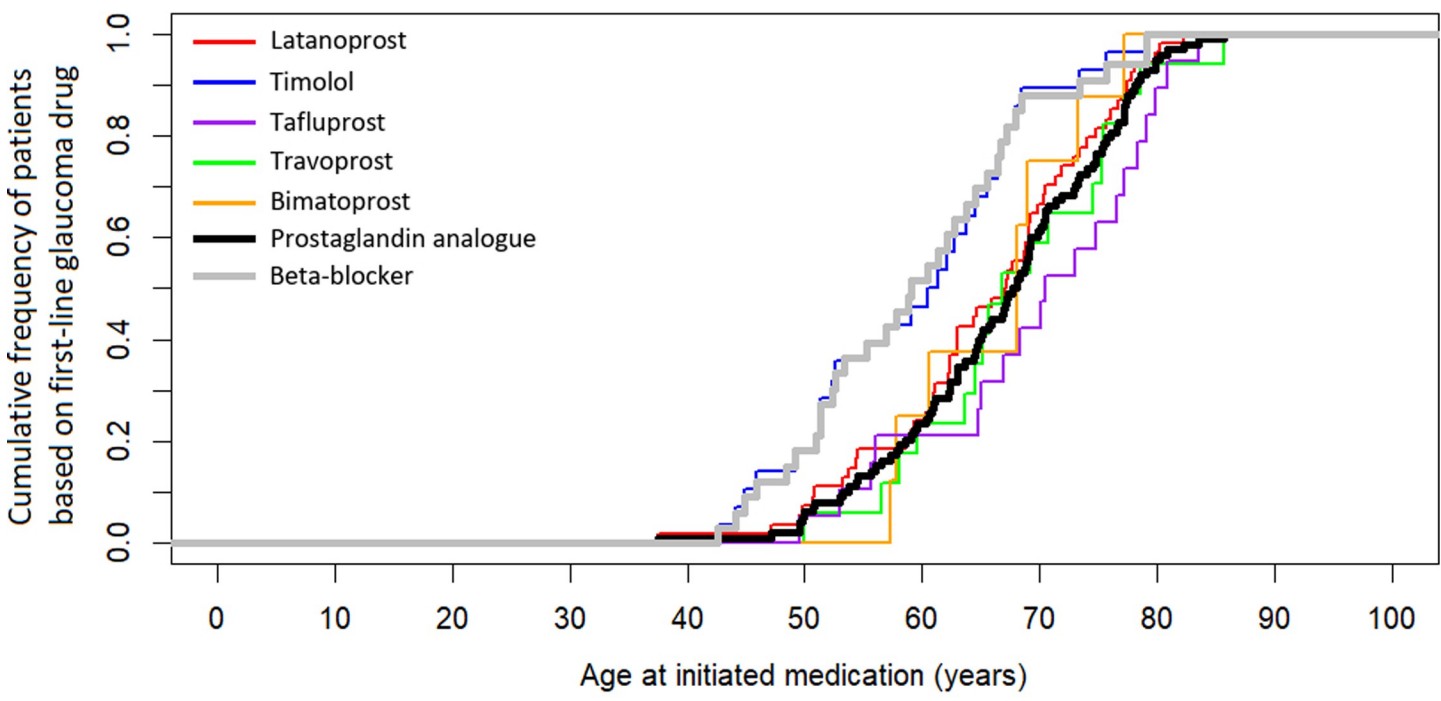

**Fig 5. Mean age at initiation of therapy of the two most common first-line glaucoma monotherapy drug classes and the five most common drug types during the follow-up between 1995 and 2019 in the FinHealth 2017 Survey sample representing the Finnish adult population aged 30 years or older.**

## Discussion

Due to many individual-based factors that need to be accounted for in glaucoma therapy, glaucoma patients usually undergo many treatment modifications. The knowledge of these is limited due to lack of population-based data and extensive follow-up periods. A detailed investigation of glaucoma medication patterns would provide valuable information on patient behavior and used glaucoma therapies for the healthcare systems to adjust and improve glaucoma care, particularly as the demand for glaucoma care is expected to increase due to the ageing population and the increasing life expectancy. To fill this gap in knowledge, we investigated the patterns of first-line glaucoma drug monotherapies in Finland during a 25-year follow-up period based on a population-based cohort. This is to the best of our knowledge the longest follow-up which has been used in determining these patterns. By providing time trends in glaucoma management during the recent decades, the current study can aid in planning how to best meet the increasing glaucoma care needs in the future.

Our main finding is that the decline in the continuation of the initial monotherapy occurs early in the therapy journey: the proportion of patients continuing with their initial glaucoma drug monotherapy had declined to 64% after 1 year and it further declined to 21% after 5 years of follow-up. Non-persistence is still a common and consistent problem with over one third of patients having significant delays in refills during their first-line glaucoma monotherapy. However, the length of use of the first-line glaucoma monotherapy of over 5 years has prolonged during the shift from beta-blockers to prostaglandin analogues as the prominent first-line glaucoma drug in the 2000s. Timolol fixed-combinations were the most prominent second-line glaucoma therapy with a share of 39% after 5 years of follow-up. The results of this study demonstrate the individual nature of glaucoma and indicate the necessity of careful follow-up of the glaucoma patients, individual tailoring of glaucoma therapy, and broader use of

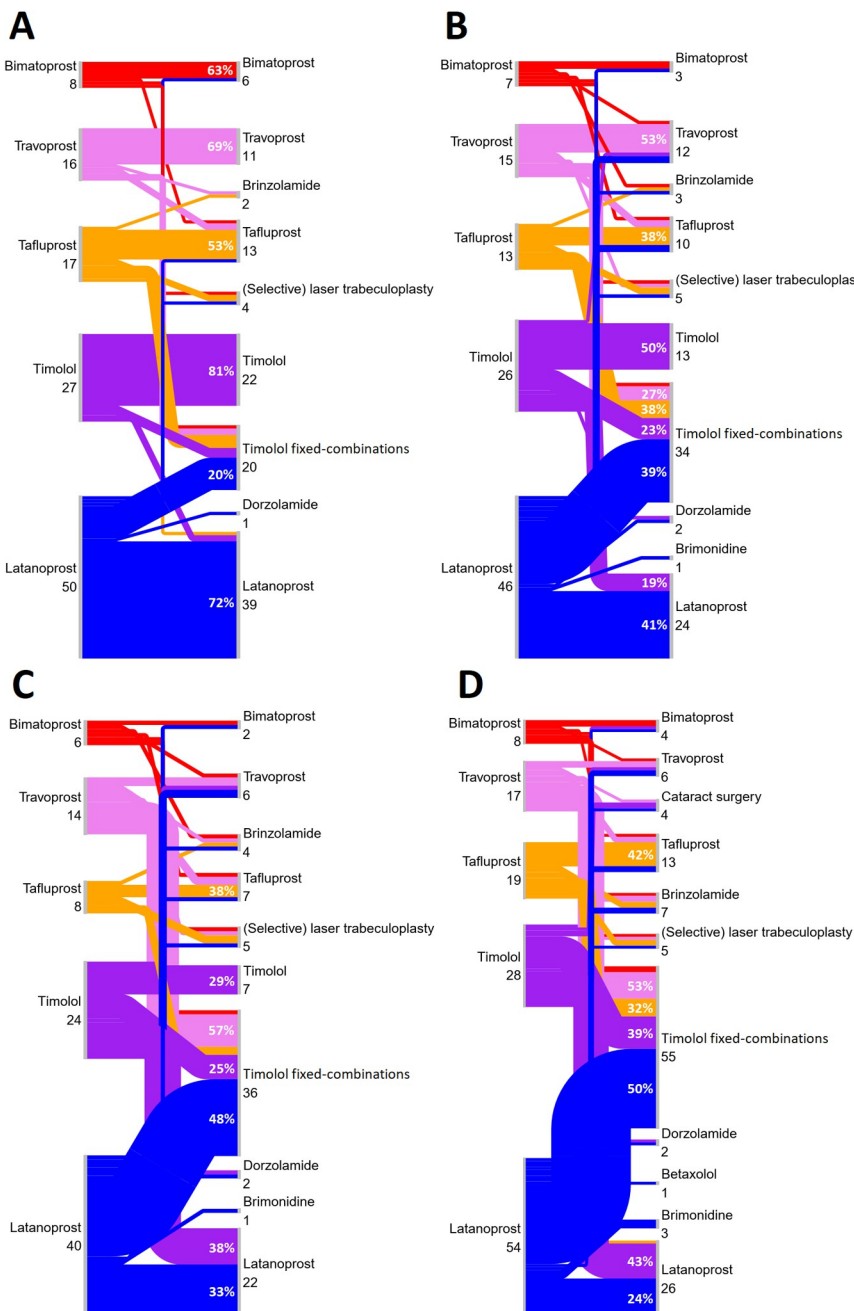

**Fig 6.** Sankey diagram illustrating the number of glaucoma patients treated with the five most common first-line glaucoma drugs and the number of patients who either had a switch to second-line therapy or who had no change to their therapy during (A) 1 year of follow-up, (B) 3 years of follow-up, (C) 5 years of follow-up, and (D) total follow-up during 1995–2019 in the FinHealth 2017 Survey sample representing the Finnish adult population aged 30 years or older. Patients with shorter follow-up per respective follow-up period (A–C) were excluded. All patients were included in the total follow-up (D). The percentages of the most common switches or continuations in relation to the first-line glaucoma drugs are also shown.

therapies like selective laser trabeculoplasty, however, not forgetting the individual efficacy of them.

In our study, two-thirds, one third and one fifth of patients continued with their first-line glaucoma monotherapy after 1, 3, and 5 years of follow-up, respectively. This is noticeably lower than in a previous healthcare insurance claims database study conducted in the US during 2010–2014, in which 56% of newly diagnosed glaucoma patients continued with their initial drug monotherapy during the first 4 years of treatment [13]. On the other hand, based on a systematic literature review, 40% continued with their initial therapy after 1 year from therapy start without discontinuation or change of therapy [22], which is lower than in our study; however, their percentage is based on a mean from 12 studies, in which the percentage individually ranged from 14% to 67% [22]. These differences could be explained by variations in populations as well as local factors such as availability of drugs, practices in reimbursements, and guidelines. In addition, the high prevalence of exfoliative glaucoma in Finland could be a factor in the need to move from the initial therapy sooner.

Although no significant difference in the length of use was observed between glaucoma drug classes and types during the first 5 years of treatment in our study, among those continuing with their first-line treatment after 5 years of follow-up prostaglandin analogues were associated with a significantly longer length of use in comparison to beta-blockers. This is probably due to the better efficacy of the prostaglandin analogues and the more easy once a day dosing. In line with our results, Goldshtein et al. reported a noticeably longer median time till discontinuation or switching among first-line prostaglandins (3.6 years) compared with other drug classes based a population-based study conducted in Israel in 2003–2012 [11]. However, they observed a median time of 1.8 years for first-line beta-blockers, whereas in our study the mean time of continuation was 3.5 years. In addition to differences in populations and local factors, this could be explained by the follow-up period in our study starting earlier, in the 1990s when beta-blockers were more commonly used as a first-line glaucoma monotherapy. Interestingly, Diestelhorst et al. reported that as early as in 1996–1998 patients who initially received latanoprost stayed on therapy more than twice as long as did those treated first with a beta-blocker during a 2-year follow-up based on a multicenter medical chart review study conducted in multiple countries [23].

Prostaglandin analogues, latanoprost in particular, were the most prominent first-line glaucoma monotherapy drug during the total 25-year follow-up in our study. After the introduction of latanoprost in Finland in 1997 its use increased quickly, and from 2003 onwards prostaglandin analogues have remained as the most used first-line glaucoma drug class in Finland. In similar, prostaglandins have also been reported in many other studies as the prominent prescribed medical therapy in the 2000s followed by beta-blockers [12,24,25]. Interestingly, based on a population-based study conducted in Israel in 2003–2012, the most common first-line drug class for glaucoma were beta-blockers with a share of 45% followed by prostaglandins with 27% [11].

The non-persistence regarding first-line glaucoma monotherapies can be considered high in our study: over one third of all patients had one or more over 1-month delays between 3-month refills during their initial therapy. The rate of non-persistence has been reported higher in previous studies [26–29], although the results may not be directly compared due to differences in the length of delay and follow-up as well as the inclusion of other than first-line monotherapies. Furthermore, in Finland glaucoma medication is reimbursed 100% by KELA, which evidently is improving the persistence rate. The persistence was similar between prostaglandin analogues and beta-blockers in our study, whereas in previous studies prostaglandin analogues have been reported having higher persistence rates than other drug classes [23,26,27]. Interestingly, in our study, the proportion of all delayed refills declined during the

25-year follow-up while the proportion of the most severe delays somewhat increased. The general decline in delayed refills could be explained by the increased recognition of the importance of regularly administered medication, whereas the increase in most severe delays likely reflects the ageing of the study cohort, considering the increasing possibility of forgetfulness due to memory disturbances, dementia, and other conditions.

In our study, approximately one fifth of glaucoma patients treated with medication had their therapy initiated with multiple antiglaucoma agents. This figure is in line with a population-based study conducted in Israel in 2003–2012, in which 22% of glaucoma patients treated with medication had either a fixed-combination drug or several separate drugs with multitherapy as a first-line glaucoma therapy [11]. This behavior is contradictory to the glaucoma guideline [30], and the reason for it is not known. If initiated with multitherapy, it is difficult to observe which agents are effective, leading to the possibility of inefficient treatment with increased risk of side-effects. Therefore, it is important to improve the awareness of the guidelines and adherence to them.

Timolol fixed-combinations were the most common second-line therapy according to our data with its share increasing from 17% to 39% after 1 and 5 years of follow-up, respectively, and its total share was 44% of all second-line therapies. Its use as a second-line therapy increased noticeably between 1995 and 2019. In relation to our results, based on a population-based data from 2003 to 2017 in Australia the use of fixed-combination medications showed a 9.2-fold increase between the two time points [24].

A 4.6-fold increase in all cases of laser trabeculoplasty was observed between 2003 and 2017 in Australia [24]. No similar trend was observed in our study. Although selective laser trabeculoplasty has been suggested as a safe and effective first-line alternative to glaucoma treatment, is not routinely offered as first-line treatment [31], which probably explains the absence of visible trends in laser operations in our study. Furthermore, our data has limited coverage of the selective laser trabeculoplasty in private sector, which is likely to further reduce the number of laser operations in our dataset.

The main strength of our study was the use of data from a nationwide survey linked to comprehensive register data follow-up for 25 years. The survey sample represents the Finnish adult population in 2017, and the survey addressed public health issues more comprehensively than national health surveys do on average. In addition, the study population did not consist of specific patient groups collected from health-care units allowing for better generalization of the results. The participation rate in the survey was relatively high and we further corrected it by applying the weights. We conducted all analyses on the largest possible number of participants. Details of prescription were available for each purchase allowing the determination of the precise expected refill date.

Our study also has limitations. We excluded patients who started on combination therapy, which may affect the representativeness of the findings and make them most applicable to patients initiating monotherapy. The cohort was selected at a single time point, and as such, the distribution of initial glaucoma therapies per year does not represent the distribution at population level each year. Similarly, comparing the age at initiation of therapy between glaucoma drug classes and types is biased at the early years of the follow-up due to the younger age of the cohort. As such, the significantly younger age at initiation of timolol monotherapies should be interpreted with caution. The survey included predominantly participants of Finnish origin; therefore, the results may not be directly applicable to other countries. In addition, interpretation of the differences between our results and those of previous studies can be rather difficult due to the heterogeneity of methods for measuring continuation and persistence and the differences in patient selection criteria.

In conclusion, the decline in the continuation of the initial glaucoma monotherapy occurs early in the therapy journey, although the length of use has prolonged among glaucoma patients who have continued with first-line prostaglandin analogues for more than 5 years. The non-persistence to first-line glaucoma monotherapies remains a significant issue. There is a need for education and tailoring of the therapeutic scheme particularly for glaucoma patients at the start of the therapy journey and monitoring the treatment scheme of all glaucoma patients during the follow-up.

## Supporting information

**S1 Table. Percentages of different glaucoma diagnoses among glaucoma patients with drug monotherapy as a first-line glaucoma therapy in the FinHealth 2017 Survey sample.** (DOCX)

## Author Contributions

**Conceptualization:** Hannu M. T. Uusitalo.

**Formal analysis:** Petri K. M. Purola.

**Resources:** Seppo V. P. Koskinen.

**Supervision:** Hannu M. T. Uusitalo.

**Visualization:** Petri K. M. Purola.

**Writing – original draft:** Petri K. M. Purola.

**Writing – review & editing:** Petri K. M. Purola, Seppo V. P. Koskinen, Hannu M. T. Uusitalo.

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
