## [Decision Letter · Decision Letter 0]

8 Nov 2024

PONE-D-24-25991First-line glaucoma monotherapy medication patterns in Finland during 1995–2019 based on a population-based studyPLOS ONE

Dear Dr. Purola,

Thank you for submitting your manuscript to PLOS ONE. After careful consideration, we feel that it has merit but does not fully meet PLOS ONE’s publication criteria as it currently stands. Therefore, we invite you to submit a revised version of the manuscript that addresses the points raised during the review process.

We look forward to receiving your revised manuscript.

Kind regards,

Andrzej Grzybowski

Academic Editor

PLOS ONE

Journal requirements: When submitting your revision, we need you to address these additional requirements. 1. Please ensure that your manuscript meets PLOS ONE's style requirements, including those for file naming. The PLOS ONE style templates can be found at https://journals.plos.org/plosone/s/file?id=wjVg/PLOSOne_formatting_sample_main_body.pdf and https://journals.plos.org/plosone/s/file?id=ba62/PLOSOne_formatting_sample_title_authors_affiliations.pdf 2. We note that you have indicated that there are restrictions to data sharing for this study. For studies involving human research participant data or other sensitive data, we encourage authors to share de-identified or anonymized data. However, when data cannot be publicly shared for ethical reasons, we allow authors to make their data sets available upon request. For information on unacceptable data access restrictions, please see http://journals.plos.org/plosone/s/data-availability#loc-unacceptable-data-access-restrictions.  Before we proceed with your manuscript, please address the following prompts: a) If there are ethical or legal restrictions on sharing a de-identified data set, please explain them in detail (e.g., data contain potentially identifying or sensitive patient information, data are owned by a third-party organization, etc.) and who has imposed them (e.g., a Research Ethics Committee or Institutional Review Board, etc.). Please also provide contact information for a data access committee, ethics committee, or other institutional body to which data requests may be sent. b) If there are no restrictions, please upload the minimal anonymized data set necessary to replicate your study findings to a stable, public repository and provide us with the relevant URLs, DOIs, or accession numbers. Please see http://www.bmj.com/content/340/bmj.c181.long for guidelines on how to de-identify and prepare clinical data for publication. For a list of recommended repositories, please see https://journals.plos.org/plosone/s/recommended-repositories. You also have the option of uploading the data as Supporting Information files, but we would recommend depositing data directly to a data repository if possible. Please update your Data Availability statement in the submission form accordingly. 3. Please include your full ethics statement in the ‘Methods’ section of your manuscript file. In your statement, please include the full name of the IRB or ethics committee who approved or waived your study, as well as whether or not you obtained informed written or verbal consent. If consent was waived for your study, please include this information in your statement as well. 

Reviewers' comments:

Reviewer's Responses to Questions

**Comments to the Author**

1. Is the manuscript technically sound, and do the data support the conclusions?

Reviewer #1: Yes

Reviewer #2: Yes

Reviewer #3: Yes

Reviewer #4: Yes

Reviewer #5: Yes

2. Has the statistical analysis been performed appropriately and rigorously? 

Reviewer #1: Yes

Reviewer #2: Yes

Reviewer #3: Yes

Reviewer #4: Yes

Reviewer #5: Yes

3. Have the authors made all data underlying the findings in their manuscript fully available?

Reviewer #1: No

Reviewer #2: Yes

Reviewer #3: No

Reviewer #4: No

Reviewer #5: Yes

4. Is the manuscript presented in an intelligible fashion and written in standard English?

Reviewer #1: Yes

Reviewer #2: Yes

Reviewer #3: Yes

Reviewer #4: Yes

Reviewer #5: Yes

5. Review Comments to the Author

Reviewer #1: As a reviewer for this article, I would like to express my overall positive evaluation of the study. The research, which analyzes the patterns of first-line glaucoma monotherapy in Finland from 1995 to 2019, undoubtedly involves a significant amount of work. It provides important insights into the treatment trends for glaucoma patients. The findings regarding the shift from beta-blockers to prostaglandin analogs, as well as the declining persistence rates among patients. At the same time, I was impressed by Table 2, Fig3, and Fig6. I was pleasantly surprised by the series of issues reflected through the indicator of delayed refills. Although the article focuses on the situation in Finland, I believe these results hold significant relevance for global clinical practice and future research. I appreciate the substantial contributions made by the author team in this study.

However, I believe some minor revisions are necessary before publication.

1. The subheadings of various sections in the article are of similar boldness and size. I recommend redistributing them to facilitate readers' immediate understanding of the overall structure of the article.

2. While the references provided by the authors are indeed well-targeted and authoritative, and they appear in their appropriate locations, it is worth noting that many of them are over 5 years old, with some reaching 10 years or even 20 years. In particular, references 5-8, which aim to showcase the research on medication adherence among glaucoma patients and highlight the importance of adherence in disease control, are review articles from 13-16 years ago. If possible, replacing these with more recent literature could enhance the persuasiveness of the article's arguments.

3. In the discussion section, a deeper exploration of the potential factors leading to changes in medication usage patterns could be beneficial, such as patients' subjective experiences, changes in healthcare policies, and the impact of patient education. The article frequently mentions that prostaglandins rapidly replaced beta-blockers; however, we know that prostaglandins are typically administered once daily. Could you further analyze whether the increased adherence is due to the reduced frequency of medication or fewer adverse effects associated with the drugs? I believe such analysis and outlook would provide meaningful guidance for clinical practice.

4. In line 340, you mention that the first-line medication situation in Finland differs from that in Israel. As this article is set to be published in a global open-access journal, it would be beneficial to provide an explanation for this difference to enhance its international relevance. A similar situation arises in line 308 regarding the potential reasons for better medication adherence among glaucoma patients in the United States. Addressing these points would significantly improve the article.

Reviewer #2: In the introduction, there is a need for the disease "glaucoma" to be well defined.

Minor grammatical edits is required and highlighted.

The manuscript describes a technically sound scientific research with data that supports the conclusion.

The statistical analysis is sound and rigorous with good illustrations.

Results were clearly presented. However, there are a lot of repetition of results in the discussion.

Reviewer #3: This is a very interesting study created with an amount of longitudinal linked data that is rarely available. Nonetheless, the small number of patients included in the analysis is something to be considered (n=141).

Lines 55 to 57 – laser is not only a first line therapy for those specific cases. It has a much broader indication as first-line therapy

Very good methodology.

Results – would mention the prevalence of glaucoma patients (as per this methodology) in the population sample - +- 2.2% which matches other studies from European countries. (eg https://pubmed.ncbi.nlm.nih.gov/28368997/)

Figure 3 – what is the unit of the y axis?

Figure 3 – what do you believe is the reason for the PGA trend shape? Increasing until 2014 and then decreasing?

Would maybe say something at the end of results regarding the frequency of switching to another monotherapy PGA versus switching to a fixed combination with timolol.

Discussion

Lines 307 to 315 – could the amount of PXF glaucoma in Finland (versus almost anywhere else) be a factor in the need to move from the initial therapy sooner?

Would it make sense to discuss the possibility for tachyphylaxis with betablockers?

I understand that the lower mean age with the beta blockers may indeed be biased, but could this also be because older patients may have more pathology that contra indicates betablockers and also the avoidance of PGA in young people since they can lead to PGA-related orbitopathy?

Reviewer #4: Very interesting research. Please expand on the possible reasons/implications of these findings in order to appeal to clinicians/a wider audience who may be interested in policy changes or intervention planning

Reviewer #5: This study investigates long-term patterns in first-line glaucoma monotherapy in Finland over a 25-year period, providing insights into treatment trends and persistence rates. It reveals an early decline in therapy continuation and a shift from beta-blockers to prostaglandin analogues, underscoring ongoing challenges in maintaining adherence to glaucoma treatment

• In the methods section: "The authors chose non-parametric tests (Mann–Whitney U and Kruskal–Wallis) due to data skewness. For added clarity, the authors might specify which variables were analyzed with each test, as this context would strengthen the rationale behind the analysis choices.

• In the discussion: the authors reference the same study from Israel (2003–2012) multiple times, which may create the impression of separate studies. Combining all comparisons with this study into a single paragraph would improve clarity and coherence.

• In the discussion: While the conclusion is strong, the authors could further highlight the broader implications of the findings and suggest practical pathways for enhancing glaucoma care. They might consider adding a brief note on future research directions, such as exploring factors influencing non-persistence, assessing the impact of newer treatment options on long-term outcomes, or evaluating patient education strategies to improve adherence

• In the discussion: The limitations are valid. To further strengthen this section, consider noting the potential impact of excluding patients who started on combination therapy, as this may affect the representativeness of the findings, making them most applicable to patients initiating monotherapy.

• In the discussion: The study reveals a significant early decline in the continuation of first-line monotherapy, as well as a shift from beta-blockers to prostaglandin analogues as the preferred first-line therapy. Notably, prostaglandin analogues showed greater persistence beyond five years compared to beta-blockers. Expanding the discussion to explore possible reasons for both the early decline and this therapeutic shift—such as factors affecting adherence, improved intraocular pressure (IOP) control, convenience, or fewer side effects—could provide deeper insights into the practical implications for glaucoma management.

• "The authors could consider revising the text to use active voice, as this adjustment may enhance the clarity and engagement of the writing."

6. PLOS authors have the option to publish the peer review history of their article (what does this mean?). If published, this will include your full peer review and any attached files.

Reviewer #1: **Yes: **Bin Lin

Reviewer #2: **Yes: **Ojo Perpetua ODUGBO

Reviewer #3: No

Reviewer #4: No

Reviewer #5: No

---

## [Author Response · Author response to Decision Letter 0]

16 Dec 2024

Dear Editor,

We would like to thank you and the reviewers for the valuable comments. We have carefully analyzed all the comments and responded accordingly. Please find below the detailed list of the changes we have made and the answers and comments to the reviewers’ remarks.

Revisions for Editor:

We have rechecked the style req

uirements as suggested.

We have updated the Data Availability statement accordingly:

“The data from FinHealth 2017 and its precursors Health 2000 and Health 2011 are not publicly available as they include confidential information that could compromise the privacy of the participants. The data can be used for research and monitoring of health, wellbeing, functioning, and use of services of the population at THL and with collaborators based on collaboration agreement (more information: terveys-2000-2011@thl.fi). The data available from the THL Biobank cover those who have participated in the health examination and have given consent to biobanking and can be applied via the THL Biobank in accordance with the biobank act and THL Biobank research areas (thl.fi/biobank). The Finnish Social and Health Data Permit Authority Findata may grant permits in accordance with the act on the secondary use of social and health data in Finland (www.findata.fi/en). Further inquiries can be directed to the email address terveys-2000-2011@thl.fi.”

We have now included the full ethics statement as suggested: “The FinHealth 2017 Survey was approved by the Coordinating Ethics Committee at the Hospital District of Helsinki and Uusimaa with reference 37/13/02/00/2016. All participants received an information letter regarding the study beforehand. Written informed consent was obtained from all participants.”

We have reviewed the reference list as suggested. We added three new references as suggested by Reviewer #1:

Quaranta, L., Novella, A., Tettamanti, M., Pasina, L., Weinreb, R. N., & Nobili, A. (2023). Adherence and persistence to medical therapy in glaucoma: An overview. Ophthalmology and therapy, 12(5), 2227-2240.

Tse, A. P., Shah, M., Jamal, N., & Shaikh, A. (2016). Glaucoma treatment adherence at a United Kingdom general practice. Eye, 30(8), 1118-1122.

Menino, J., Camacho, P., & Coelho, A. (2024). Persistence with medical glaucoma therapy in newly diagnosed patients. Medical Hypothesis, Discovery and Innovation in Ophthalmology, 13(2), 63.

Revisions for Reviewer #1:

1. The subheadings of various sections in the article are of similar boldness and size. I recommend redistributing them to facilitate readers' immediate understanding of the overall structure of the article.

We have now italicized the secondary headings for better clarity as suggested.

2. While the references provided by the authors are indeed well-targeted and authoritative, and they appear in their appropriate locations, it is worth noting that many of them are over 5 years old, with some reaching 10 years or even 20 years. In particular, references 5-8, which aim to showcase the research on medication adherence among glaucoma patients and highlight the importance of adherence in disease control, are review articles from 13-16 years ago. If possible, replacing these with more recent literature could enhance the persuasiveness of the article's arguments.

We thank the Reviewer for the suggestion. We have now added the following references:

Quaranta, L., Novella, A., Tettamanti, M., Pasina, L., Weinreb, R. N., & Nobili, A. (2023). Adherence and persistence to medical therapy in glaucoma: An overview. Ophthalmology and therapy, 12(5), 2227-2240.

Tse, A. P., Shah, M., Jamal, N., & Shaikh, A. (2016). Glaucoma treatment adherence at a United Kingdom general practice. Eye, 30(8), 1118-1122.

Menino, J., Camacho, P., & Coelho, A. (2024). Persistence with medical glaucoma therapy in newly diagnosed patients. Medical Hypothesis, Discovery and Innovation in Ophthalmology, 13(2), 63.

3. In the discussion section, a deeper exploration of the potential factors leading to changes in medication usage patterns could be beneficial, such as patients' subjective experiences, changes in healthcare policies, and the impact of patient education. The article frequently mentions that prostaglandins rapidly replaced beta-blockers; however, we know that prostaglandins are typically administered once daily. Could you further analyze whether the increased adherence is due to the reduced frequency of medication or fewer adverse effects associated with the drugs? I believe such analysis and outlook would provide meaningful guidance for clinical practice.

The reviewer is right, and we believe such analysis could be important. Unfortunately our data do not allow to distinguish between reduced frequency of medication or fewer adverse effects associated with the drugs, as we only know that the medication has been bought based on the prescription data.

4. In line 340, you mention that the first-line medication situation in Finland differs from that in Israel. As this article is set to be published in a global open-access journal, it would be beneficial to provide an explanation for this difference to enhance its international relevance. A similar situation arises in line 308 regarding the potential reasons for better medication adherence among glaucoma patients in the United States. Addressing these points would significantly improve the article.

The Reviewer rises an important point. We have now added the following sentence: “These differences could be explained by variations in populations as well as local factors such as availability of drugs, practices in reimbursements, and guidelines.“

Revisions for Reviewer #2:

In the introduction, there is a need for the disease "glaucoma" to be well defined.

We thank the Reviewer for the suggestion. We have carefully re-read the manuscript and we believe that the first paragraph of the Introduction will give a sufficient description of glaucoma.

Minor grammatical edits is required and highlighted.

We thank the reviewer for pointing out these errors, and we have corrected them as suggested.

This is the abbreviation for Light Amplification by Stimulated Emission of Radiation. The abbreviation is better written in captial letters.

In most glaucoma literature, laser is written lowercase, and as such we have left it as it is. For example, see the EGS glaucoma guidelines (ISBN 978-88-98320-39-4).

“3 had only a single glaucoma medication prescription during the follow-up” Why were these 3 excluded since they had monotherapy? Was monotherapy instituted on diagnosis (baseline)? Does a single glaucoma medication mean the drug was applied to only one eye? Please clarify.

The 3 subjects were excluded as they only had a single glaucoma medication prescription during the follow-up, indicating that they did not have a continuous medical treatment of glaucoma and did either have other condition than glaucoma or other type of therapy.

Results were clearly presented. However, there are a lot of repetition of results in the discussion.

We thank the Reviewer for the suggestion. We have now simplified the results in the Discussion, e.g., by omitting the repetition of the percentages.

Revisions for Reviewer #3:

This is a very interesting study created with an amount of longitudinal linked data that is rarely available. Nonetheless, the small number of patients included in the analysis is something to be considered (n=141).

We thank the Reviewer for rising an important point. The small number of patients is due to the nature of population-based studies aiming to have a nationally representative sample. In this study the original sample size of the survey was 10247 Finnish adults aged 18 years or older which should regarded as a large one. We have taken the sample size into account by utilizing population weights and calculating confidence intervals.

Lines 55 to 57 – laser is not only a first line therapy for those specific cases. It has a much broader indication as first-line therapy

Thank you for the comment. We have now revised the sentence: “Alternatively, laser treatments such as laser trabeculoplasty and laser iridotomy or surgery can be offered as a first-line glaucoma therapy, in patients with drug-related adverse reactions, in patients with difficulties in administering topical medication, and in patients with severe disease.”

Results – would mention the prevalence of glaucoma patients (as per this methodology) in the population sample - +- 2.2% which matches other studies from European countries. (eg https://pubmed.ncbi.nlm.nih.gov/28368997/)

The prevalence of register-based glaucoma was 3.1%, and the prevalence of glaucoma treated with first-line drug monotherapy 2.0% in the population 30 years or older. We have added these figures to the Results, first paragraph.

Figure 3 – what is the unit of the y axis?

The unit of the y axis is the mean n of a 3-year central moving average, which we have now added to the figure.

Figure 3 – what do you believe is the reason for the PGA trend shape? Increasing until 2014 and then decreasing?

Considering that the first Finnish Glaucoma Guidelines was published in 1999 with consequently increased awareness of the disease, it is likely that there was a surge of newly diagnosed glaucoma patients at the time and a subsequent decrease in new diagnoses. We this believe could explain the decreasing trend of the PGA after 2014 considering the decreasing number of new glaucoma cases in the FinHealth 2017 survey sample as seen in the example figure below.

Figure. Dates of first glaucoma-related diagnoses and special reimbursements among the 141 analyzed glaucoma patients with initiated monotherapy

Would maybe say something at the end of results regarding the frequency of switching to another monotherapy PGA versus switching to a fixed combination with timolol.

The frequency of switching has been presented in Figure 6. We have now added the switching frequencies: 

”During 1 year of follow-up, 9% switched to another monotherapy, 17% switched to Timolol fixed-combinations, and 3% switched to (selective) laser trabeculoplasty. During 3 years of follow-up, 20% switched to another monotherapy, 32% switched to Timolol fixed-combinations, and 5% switched to (selective) laser trabeculoplasty. During 5 years of follow-up, 27% switched to another monotherapy, 39% switched to Timolol fixed-combinations, and 5% switched to (selective) laser trabeculoplasty. And during the total follow-up, 29% switched to another monotherapy, 44% switched to Timolol fixed-combinations, and 4% switched to (selective) laser trabeculoplasty.”

Discussion

Lines 307 to 315 – could the amount of PXF glaucoma in Finland (versus almost anywhere else) be a factor in the need to move from the initial therapy sooner?

We agree the with the suggestion of the Reviewer, and we have now added the higher prevalence of exfoliative glaucoma being a possible explanation: “In addition, the high prevalence of exfoliative glaucoma in Finland could be a factor in the need to move from the initial therapy sooner.”

Would it make sense to discuss the possibility for tachyphylaxis with betablockers?

This would be interesting, but our data are not allowing us for detailed discussion. In addition to decreasing efficacy of betablocker treatment, the reason for withdrawal of betablockers can be, for example, systemic or local side-effects. Unfortunately our data are not allowing to differentiate between them.

I understand that the lower mean age with the beta blockers may indeed be biased, but could this also be because older patients may have more pathology that contra indicates betablockers and also the avoidance of PGA in young people since they can lead to PGA-related orbitopathy?

We thank the Reviewer for rising an interesting point. However, the changes in the availability of various glaucoma drugs during the follow-up period are overriding these possible explanations and were therefore very difficult to study.

Revisions for Reviewer #4:

Very interesting research. Please expand on the possible reasons/implications of these findings in order to appeal to clinicians/a wider audience who may be interested in policy changes or intervention planning

We thank the Reviewer for the comment. We have added the following points to the second paragraph in Discussion: “The results of this study demonstrate the individual nature of glaucoma and indicate the necessity of careful follow-up of the glaucoma patients, individual tailoring of glaucoma therapy, and broader use of therapies like selective laser trabeculoplasty, however, not forgetting the individual efficacy of them.”

Revisions for Reviewer #5:

• In the methods section: "The authors chose non-parametric tests (Mann–Whitney U and Kruskal–Wallis) due to data skewness. For added clarity, the authors might specify which variables were analyzed with each test, as this context would strengthen the rationale behind the analysis choices.

We have rephrased the sentence: “Therefore, Mann–Whitney U test was used for ana

---

## [Editor Report · Decision Letter 1]

18 Dec 2024

First-line glaucoma monotherapy medication patterns in Finland during 1995–2019 based on a population-based study

PONE-D-24-25991R1

Dear Dr. Purola,

We’re pleased to inform you that your manuscript has been judged scientifically suitable for publication and will be formally accepted for publication once it meets all outstanding technical requirements.

Kind regards,

Andrzej Grzybowski

Academic Editor

PLOS ONE
---

## [Editor Report · Acceptance letter]

23 Dec 2024

PONE-D-24-25991R1 

PLOS ONE

Dear Dr. Purola, 

I'm pleased to inform you that your manuscript has been deemed suitable for publication in PLOS ONE. Congratulations! Your manuscript is now being handed over to our production team.

Kind regards, 

on behalf of

Dr. Andrzej Grzybowski 

Academic Editor

PLOS ONE